# Role of Radiation Therapy in Mortality among Adolescents and Young Adults with Lymphoma: Differences According to Cause of Death

**DOI:** 10.3390/cancers14205067

**Published:** 2022-10-16

**Authors:** Xuejiao Yin, Liangshun You, Xuelian Hu

**Affiliations:** 1Department of Hematology, The First Affiliated Hospital, Zhejiang University School of Medicine, #79 Qingchun Road, Hangzhou 310003, China; 2Zhejiang Province Key Laboratory of Hematology Oncology Diagnosis and Treatment, Hangzhou 310003, China

**Keywords:** AYA, radiation therapy, lymphoma, standardized mortality ratios, causes of death, second malignant neoplasm, noncancer death

## Abstract

**Simple Summary:**

Despite its efficacy, emerging concerns exist regarding radiation therapy (RT)-associated toxicity in adolescent and young adult (AYA) lymphoma patients. However, the most current research studying the association between RT and outcome is based only on groups of lymphoma patients and is not compared with the general population, which could more accurately reveal the impact of RT on outcomes. This population-based analysis showed that after adjusting for potential confounders, RT administration is associated with a low risk of cause-specific mortality, including death due to the original diagnosis, second malignant neoplasms (SMNs), and noncancer causes, among AYA patients with lymphoma. This analysis may play a guiding role in the design of future lymphoma studies and the formulation of healthcare policies regarding the widespread use of RT, especially for AYA survivors.

**Abstract:**

Background: Despite its efficacy, emerging concerns exist regarding radiation therapy (RT)-associated toxicity in adolescent and young adult (AYA) lymphoma patients. Few long-term follow-up studies have examined the association between RT and outcomes. Methods: Lymphoma patients aged 15–39 years were identified in the Surveillance, Epidemiology and End Results (SEER) database from 1992 to 2016. Mortality was assessed by comparing those with and without RT using the Fine–Gray competing risk model. Standardized mortality ratios (SMRs) were used to assess the relative risk of death compared with the general U.S. population. Results: In total, 29,686 patients were included; 10,708 (36.07%) received RT. Cause-specific mortality was compared between patients with and without RT while considering other competing events, including death due to index cancer, second malignant neoplasms (SMNs), and noncancer causes. Patients with RT had a lower probability of death and crude 5-year cumulative incidence of death. Moreover, there were significantly lower SMRs in patients with RT than in patients without RT. Differences between the two groups were greatest for mortality due to hematological malignancies and infections. Additionally, in the RT cohort, the SMR for index-cancer-related death was highest in the first year after diagnosis and gradually decreased. Hematological malignancies and infections were the most common specific SMN and noncancer causes of death, respectively. Conclusions: RT did not increase mortality from index cancer, SMNs, or noncancer causes in AYA patients with lymphoid malignancies. The current analysis may serve as a reference for healthcare providers monitoring RT application for AYA lymphoid malignancy survivors.

## 1. Introduction

Adolescents and young adults (AYAs) aged 15 to 39 years constitute a unique group that is experiencing a faster increase in cancer incidence and slower improvement in survival rate than either older adults or younger children [1]; approximately 70,000 new cases of cancer are diagnosed every year in the USA [1]. The most common cancer in AYA patients is lymphoma, accounting for 20% of cancer cases [2].

Radiation therapy (RT) is one of the most effective treatments available for lymphomas. Although radiotherapy is being gradually replaced by other methods of treatment involving chemotherapy and immunochemotherapy, it is still a highly significant treatment. RT is beneficial to initial bulky disease or extralymphatic involvement in patients receiving modern immunochemotherapy [3]. However, a variety of complications associated with RT have been reported among survivors, even decades after therapy [4,5,6,7,8]. The notable significant side effects associated with RT can lead to increased mortality rates, especially death due to cardiovascular disease and secondary malignant neoplasms (SMNs) [9,10]. The toxicity has limited the use of RT. However, modern imaging and highly conformal RT techniques have expanded indications and reduced side effects, leading to the achievement of larger and anatomically more difficult target regions with much less radiation to normal tissues [10,11]. Furthermore, the doses of radiation have been reduced according to several large, randomized trials [12,13,14].

The association between RT and outcome was only explored among survivors of AYA lymphoma in most studies, and little is known about mortality due to RT-associated toxicities compared with the general population, which could more accurately reveal the impact of RT on outcomes. As increasing age is a natural driver of the incidence of cancer, cardiovascular diseases, infection, and mortality among the general population, this comparison with the general population is crucial to determine the risk potentially attributable to RT. Moreover, it is crucial for competing risks to be fully considered when performing these analyses. Therefore, we used data from the Surveillance Epidemiology and End Results (SEER) database to provide a more comprehensive and updated assessment of the effect of RT on cause-specific mortality, including death due to primary cancer, second tumor, and noncancer causes among AYAs with lymphoma compared with the general population.

## 2. Materials and Methods

### 2.1. Patient Database

Patients aged 15–39 years with lymphoma as the primary malignancy were identified from 13 SEER registries from 1992 to 2016. The SEER-13 program, covering up to 13.8% of the population in the USA, collects a variety of information regarding cancer patient demographics and diagnosis information, including the primary site, histology and stage, therapy, and survival. According to the International Classification of Diseases for Oncology, third-edition histology codes, and the World Health Organization lymphoid classification (2008), we included Hodgkin’s lymphoma (HL) and seven common NHL subtypes: diffuse large B-cell lymphoma (DLBCL), mantle cell lymphoma (MCL), Burkitt’s lymphoma (BL), marginal zone lymphoma (MZL), chronic lymphocytic leukemia/small lymphocytic lymphoma (CLL/SLL), peripheral T-cell lymphoma (PTCL), and follicular lymphoma (FL).

We excluded cases with the following features: (1) patients diagnosed only by autopsy or death certificate; (2) unknown cause of death; (3) follow-up less than 6 months; (4) RT use was unknown, or RT was recommended but was not known to have been administered; (5) chemotherapy was not performed, or the status was unknown.

### 2.2. Definition of Variables from the SEER

Clinical data regarding age at diagnosis, race, sex, Ann Arbor stage, year of diagnosis, primary site of involvement, marital status, therapeutic regimen, cause of death, and survival time were collected. Age at diagnosis was divided into 2 groups: 15–24 and 25–39. The stage was divided into early stage (I/II) and advanced stage (III/IV). Marital status was classified as married or unmarried (including never married, separated, divorced, widowed). Therapeutic regimens were grouped into radiation and no radiation. The era of diagnosis was divided into 1992–2001 and 2002–2016. Since 2002, 79% of Medicare beneficiaries have received rituximab-based regimens as their initial therapy.

### 2.3. Cause of Death Data

Information about the cause of death was identified according to the International Classification of Diseases, 10th edition (ICD-10). The causes of death were grouped into three categories: the “index-cancer-related death” group (death from the primary cancer); the “SMN-related death” group (death from secondary cancers); and the “noncancer-related death” group (death from noncancer causes).

Cardiovascular-related deaths and infection-related deaths were the two most common causes of noncancer deaths. Per the ICD-10 codes, the composite variable “cardiovascular disease (CVD)-related deaths” consisted of atherosclerosis; aortic aneurysm and dissection; heart disease; hypertension without heart disease; cerebrovascular diseases; or other diseases of the arteries, arterioles, and capillaries. The composite variable “infection-related deaths” consisted of sepsis, pneumonia, and other infectious and parasitic diseases, including HIV.

### 2.4. Statistical Analysis

Mortality was calculated and compared in the groups with and without RT therapy using the Fine–Gray competing risk model, which could calculate cause-specific hazard ratios. We also used the Fine–Gray competing risk model to crudely calculate the cumulative incidence function (CIF) to show the probability of developing primary and competing events, and Gray’s test was used to estimate the differences in CIF among subgroups. All statistical analyses were two-sided with a threshold of significance of *p* < 0.05.

Standardized mortality ratios (SMRs) were used to assess the relative risk of death compared to the general U.S. population via SEER*Stat software, in which data were matched to adjust for age, race, and sex in the U.S. population during the same era. The 95% confidence intervals (CIs) and corresponding *p*-values were used to estimate the risk of death with the Poisson exact method. The absolute excess risk (AER) was computed as follows: ([observed − expected deaths] × 10,000)/person-years at risk.

## 3. Results

### 3.1. Patient Characteristics

Table 1 shows the baseline demographic and treatment characteristics of the patients. A total of 29,686 AYA patients with lymphoma between 1992 and 2016 were included, of whom 10,708 (36.07%) received RT and 18,978 (63.93%) did not. Patients who underwent RT had a younger age; a lower stage; an earlier year of diagnosis; a larger proportion of patients with the HL subtype; and were more likely to be female, white, and married as opposed those who did not undergo RT. In the total cohort, 4295 died, including 2439 (56.79%), 961 (22.37%), and 895 (20.84%) patients who died due to index cancer, SMNs, and noncancer causes, respectively. A smaller proportion of patients who underwent RT died (11.23% vs. 16.29%), which was attributed to fewer patients dying due to SMNs (18.45% vs. 23.9%).

### 3.2. Competing Risk Analyses

The cause-specific mortality was compared by competing risk analysis between patients with RT and those without RT for other competing events, including death due to index cancer, SMNs, and noncancer causes, which all showed that patients with RT had a lower probability of death than those without RT (Figure 1, Figure 2, and Appendix A). The 5-year cumulative incidences of death from the index cancer, SMNs, and noncancer causes were 5.75%, 1.52%, and 0.69%, respectively, in patients treated with RT and 8.39%, 3.30%, and 1.69%, respectively, in patients not treated with RT (*p* < 0.001).

After adjusting for potential confounders, treatment with RT was significantly associated with decreased probabilities of death due to the index cancer (adjusted hazard ratio (AHR), 0.89; 95% CI: 0.81–0.97; *p* = 0.012), SMNs (AHR, 0.74; 95% CI: 0.63–0.88; *p* < 0.001), and noncancer causes (AHR, 0.77; 95% CI: 0.66–0.89; *p* < 0.001) (Table 2). In addition, male sex, advanced stage, older age at diagnosis, black race, and earlier year of diagnosis were significant predictors of an increased probability of death due to the index cancer, SMNs, and noncancer causes. Although marital status was associated with decreased SMN-specific mortality (*p* < 0.001), no difference was observed for index cancer-specific mortality (*p* = 0.31) or noncancer cause-specific mortality (*p* = 0.50). The DLBCL subtype and BL subtype were associated with higher index cancer-specific mortality and SMN-specific mortality.

### 3.3. Comparison with the General Population

#### 3.3.1. Index-Cancer-Related Deaths

Table 3 shows the comparison of the SMR for index-cancer-related death between patients with RT and without RT. The SMR for index cancer was smaller for patients with RT than for those without RT (SMR, 534.66 (95% CI, 489.9–582.41) vs. SMR, 799.25 (95% CI, 754.33–846.15)), and this was the category with the greatest difference between the two groups. Regardless of whether patients with or without RT were considered, the SMR for index-cancer-related death was highest in the first year after diagnosis (SMR, 3221.36 (95% CI, 2639.98–3892.68) in patients with RT, and SMR, 4981.89 (95% CI, 4425.88–5588.43) in patients without RT) and gradually decreased over the follow-up period. For patients with or without RT, black race (SMR, 633.48 (95% CI, 493.83–800.37) in patients with RT, and SMR, 829.34 (95% CI, 717.84–953.25) in patients without RT), advanced stage (SMR, 937.95 (95% CI, 814.81–1074.45) in patients with RT, and SMR, 1018.5 (95% CI, 946.9–1094.07) in patients without RT), and PTCL subtype (SMR, 1331.76 (95% CI, 834.61–2016.3) in patients with RT, and SMR, 1544.74 (95% CI, 1184.34–1980.28) in patients without RT) were associated with the highest SMRs for index-cancer-related death.

#### 3.3.2. SMN-Related Deaths

Table 4 and Appendix A show the comparison of the SMR for SMN-related death between patients with RT and without RT. There was a relatively lower SMR for SMN-related deaths in patients with RT than in those without RT (SMR, 3.59 (95% CI, 2.84–4.48) vs. SMR, 5.31 (95% CI, 4.54–6.18)). Differences between the two groups were the greatest for mortality due to hematological malignancies (SMR, 16.52 (95% CI, 10.35–25.01) in patients with RT vs. SMR, 45.84 (95% CI, 36.81–56.41) in patients without RT). Among the patients with RT, hematological malignancies were the most common cause of SMN-related death. Soft tissue solid tumors were the most common cause of solid SMN deaths (SMR, 11.11 (95% CI, 3.61–25.93)). The SMR was also significantly elevated for both digestive system malignancies (SMR, 2.28 (95% CI, 1.21–3.9)) and respiratory system malignancies (SMR, 2.67 (95% CI, 1.33–4.78)).

Among the patients without RT, hematological malignancies (SMR, 45.84 (95% CI, 36.81–56.41)) and bone and joint malignancies (SMR, 11.87 (95% CI, 2.45–34.68)) were the most common causes of SMN-related deaths. The highest risk of hematological malignancy-related death was observed for chronic lymphocytic leukemia (CLL) (SMR, 632.21 (95% CI, 455.64–854.57)). The third leading cause of SMN-related death was gynecologic and female genital cancers (SMR, 3.63 (95% CI, 1.57–7.16)).

#### 3.3.3. Noncancer-Related Deaths

Table 4 and Appendix A show the comparison of the SMR for noncancer-related deaths between patients with RT and without RT. There was a significantly lower SMR for noncancer-related deaths in patients with RT than in those without RT (SMR, 2.45 (95% CI, 2.15–2.77) vs. SMR, 4.56 (95% CI, 4.23–4.91)). Differences between the two groups were the greatest for infection-related mortality (SMR, 14.67 (95% CI, 12.01–17.74)) in patients with RT vs. SMR, 33.04 (95% CI, 29.85–36.48) in patients without RT). In addition, there was a relatively lower SMR for CVD-related deaths in patients with RT than in those without RT (SMR, 2.42 (95% CI, 1.82–3.16)) in patients with RT vs. SMR, 2.8 (95% CI, 2.28–3.42) in patients without RT). Among the patients with RT, the largest number of noncancer-related deaths was due to infections, with the highest risk in the first year after diagnosis (SMR, 174.66 (95% CI, 128.78–231.58)) (Appendix A). The risk due to CVD was also observed to be significantly elevated (SMR, 2.42; 95% CI, 1.82–3.16), with the greatest risk occurring more than 10 years after diagnosis (SMR, 2.89 (95% CI, 2.03–3.98)) (Appendix A).

Among the patients without RT, infection was the most common cause of noncancer-related death, with mortality decreasing over time (SMRs of 255.75, 51.11, 11.64, and 7.9 for within 1 year, 1 to 5 years, 6 to 10 years, and more than 10 years after diagnosis, respectively) (Appendix A). A significantly elevated SMR was observed for cardiovascular causes (Appendix A). Patients also had an elevated SMR for chronic obstructive pulmonary disease and allied conditions (SMR, 2.51 (95% CI, 1.01–5.16)) and nephritis, nephrotic syndrome, and nephrosis (SMR, 6.6 (95% CI, 3.29–11.81)).

## 4. Discussion

Compared to older patients, AYAs have more potential years of life that can be saved with effective treatment, and an appropriate renewed focus has been placed on prognostic factors to decrease morbidity due to treatment. Although the radiosensitivity of lymphoma has long been recognized, RT-associated SMN and toxicities are major drawbacks. However, the radiation dose, treatment fields, radiation techniques, and quality of radiation have all changed substantially in the past three decades. Using a large population-based registry, the present study provides an up-to-date comparison of cause-specific mortality, including death due to the index cancer, SMNs, and noncancer causes, between lymphoma patients with RT and without RT, with detailed evaluations stratified according to sex, age, race, follow-up time, era of diagnosis, marital status, and disease subtype. After adjusting for potential confounders, treatment with RT was significantly associated with a decreased probability of death due to the index cancer, SMNs, and noncancer causes.

We observed that treatment with RT was significantly associated with a decreased probability of death due to the index cancer, SMNs, and noncancer causes. This agrees with the findings of several previous studies, which have suggested that omission of RT was an independent prognostic factor for death [15]. Another Korean study showed that chemotherapy alone could only control lymphoma in 56% of cases requiring consolidation RT [16]. The main reason for this phenomenon is the introduction of low-dose (20- to 30-Gy) radiation and smaller treatment fields from the late 1970s through to the mid-1980s, especially in AYA patients [17,18]. This was motivated by the need to reduce musculoskeletal toxicities in the beginning, but it was later found to also reduce other toxicities, such as secondary cancers and cardiopulmonary effects. In addition, advanced imaging, clinical staging, and risk stratification have already led to the fine tuning of the intensity of therapy to match the risk [18]. Functional imaging studies, such as positron emission tomography and magnetic resonance spectroscopy combined with three-dimensional anatomical models of patients, are modern anatomical imaging technologies that yield improved imaging results. These improved imaging results can be used to more accurately ascertain the tumor volume and its spatial relationships with the surrounding tissue and organs. Taking full advantage of these improved imaging techniques, three-dimensional treatment planning systems have promoted three-dimensional conformal RT to become a standard of practice [19]. Another cause may be that RT techniques have improved considerably over the past decade [20]. The use of highly conformal radiation techniques, such as intensity-modulated radiation therapy (IMRT) and charged particle therapy (proton radiation therapy), has become increasingly common in recent years [20]. IMRT can be used to achieve even greater dose conformity due to the computer-aided optimization process. This process creates a fluence of photons per radiation beam customized to the patient [21]. 

Additionally, if RT is performed, a lower relapse rate can be achieved, which is also an important reason that treatment with RT was significantly associated with a decreased probability of death. Yahalom et al. [22] reported that patients with advanced-stage classic HL treated with RT had relatively better relapse-free survival (RFS) and overall survival (OS). The RAPID study randomized 420 patients with stages IA and IIA classic HL whose FDG/PET showed negative results after three cycles of ABVD to receive involved field RT (30 Gy) or no further therapy. The preliminary analysis after a median follow-up of 48.6 months showed similar survival in both groups (92.8% vs. 90.0%) but a trend toward less relapse in the RT arm (3.8% vs. 9.5%, respectively) [23]. Our study also confirmed that patients with advanced-stage disease had a higher SMR for death from the original diagnosis than those with early-stage disease. This was consistent with multiple studies on patterns of relapse that have shown recurrences in over 90% of patients with advanced-stage HL [24]. Once relapse occurs, the salvage therapy performed involves more aggressive therapies and even stem cell transplantation, and survival is greatly reduced [25].

We found that hematological malignancies were the most common cause of SMN-related death among the patients with RT. Mertens et al. [26] reported that mortality due to hematopoietic malignancies accounted for 27% of SMN-related mortality among childhood cancer survivors. Although not specific to survivors of lymphoma, a population-based study also indicated that mortality due to hematopoietic malignancies accounted for 21% of SMN-related mortality among AYA cancer survivors [27]. Treatment-related hematological malignancies (e.g., acute myeloid leukemia, AML) are well-documented complications of cancer treatment [28]. The pathogenic/etiological mechanisms are as yet unknown but involve the following mechanisms: (1) Ionizing radiation may result in mutations to mitochondrial and nuclear DNA, leading to chromosomal instability. Precursor and incipient neoplasms may later develop following these subclinical changes [29]. (2) Radiation may affect circulating lymphocytes and progenitor cells in the bone marrow [30].

However, it is worth noting that the SMN with the highest frequency may not equate to the SMN with the greatest risks of mortality. O’Brien et al. [31] reported that 110 children completed HL treatment; the median follow-up time was 20.6 years. A total of 18 patients developed one or more SMNS, including 4 with leukemia, 4 with sarcoma, 5 with thyroid cancer, and 6 with breast cancer. All four cases of secondary leukemia were fatal. For patients with second solid tumors, the mean (±SE) 5-year disease-free survival and OS were 76% ± 12% and 85% ± 10%, respectively, during a median follow-up of 5 years after SMN diagnosis. Moreover, the SMN fears in current clinical practice may be based on historical treatment. For example, the high risk of subsequent breast cancer after mantle field radiation therapy (MRT) has been well determined [32]. However, Conway et al. [33] showed that patients who received minor field radiation therapy (SFRT) did not have a greater risk of breast cancer than those who did not accept RT. This result is opposite to those in previous reports [32,34,35]. A related clinical trial (NCT01120353) by the Childhood Cancer Survivor Study (CCSS) was carried out that studied AYA patients exposed to specific treatment modalities, such as chemotherapy, radiation, and surgery, who have an increased risk of adverse health results.

We showed that there was a relatively lower SMR for cardiac deaths in patients with RT than in those without RT. This agrees with the findings of a previous study, which suggested that early DLBCL without RT has increased cardiac mortality compared with those with RT [36]. A combination of anthracycline-based chemotherapy, anti-CD20 antibodies, and RT is the standard treatment for early-stage DLBCL [37,38]. Therapies without RT usually use protracted and/or more intensive chemotherapy schedules [38]. Anthracycline chemotherapy is strongly related to a dose-dependent risk of left ventricular systolic dysfunction [39]. We hypothesize that the main reason for this phenomenon is that patients who did not receive RT, those who received longer cycles of chemotherapy and therefore accumulated exposure to anthracyclines, had an increased risk of cardiac death. Pugh et al. [36] observed that not receiving RT was still independently associated with an increased risk of heart-specific death in multivariate analyses (HR, 1.32; 95% CI: 1.13–1.54; *p* = 0.0005).

In the pre-rituximab era, three cycles of CHOP plus RT became the standard of care after the phase III SWOG S8736 study showed improved PFS and OS compared with 8 cycles of CHOP alone in localized intermediate- and high-grade non-Hodgkin’s lymphoma [40]. In modern clinical practice, chemotherapy along with rituximab have now emerged as the backbone of strategy for optimal systemic disease treatment, and RT is usually evolved to consolidate these systemic therapies to improve local control in large B-cell lymphoma. In patients who cannot tolerate chemotherapy, routine RT may spare chemotherapy; therefore, combined modality treatment (CMT), including abbreviated chemotherapy plus RT, remains an important treatment for some patients. In our analysis, SMN-related mortality was not obviously influenced by the additional application of rituximab since 2002.

The strengths of the current study included the large number of AYA patients with lymphoma, which allowed us to evaluate the influence of RT on cause-specific mortality, including death due to index cancer, SMNs, and noncancer causes. Because of the sheer magnitude of accessible population data, these analyses have great statistical power. Therefore, they allow for the determination of small but clinically meaningful differences between treatment subgroups that might be overlooked in smaller cohorts. Moreover, the use of the SEER population-based registry easily allowed for comparison of mortality risk in survivors of AYA lymphoma with that of the general population, and biases based on hospital-specific series were avoided.

There are some limitations to our findings with regard to the information available in the SEER database. First, no information regarding details of RT planning (e.g., the type, doses, and fields) is available in the SEER. Therefore, we could not ascertain the clinical rationale for performing RT and further assess the risk of death from doses and fields of RT. Second, there was no record of the morbidity of nonfatal SMN and the temporal pattern of incidence/risk in the present article, which is an important part of the incidence of common secondary tumors caused by radiotherapy. Third, there was a difference in the patient group divided. Thus, when comparing the factors of death, there may be bias. Finally, follow-up within the present analysis was too short to leave enough time to measure the mortality of all RT-induced SMN or cardiovascular events. This should be considered a limitation in incorporating the results of this study.

## 5. Conclusions

In summary, this population-based analysis showed that after adjusting for potential confounders, RT administration is associated with a low risk of cause-specific mortality, including death due to the original diagnosis, SMNs, and noncancer causes, among AYA patients with lymphoma. This analysis may play a guiding role in the design of future lymphoma studies and the formulation of healthcare policies regarding the widespread use of RT, especially for AYA survivors. Prospective trials are needed to further define and refine the value of RT in lymphoma patient subsets.

## Figures and Tables

**Figure 1 cancers-14-05067-f001:**
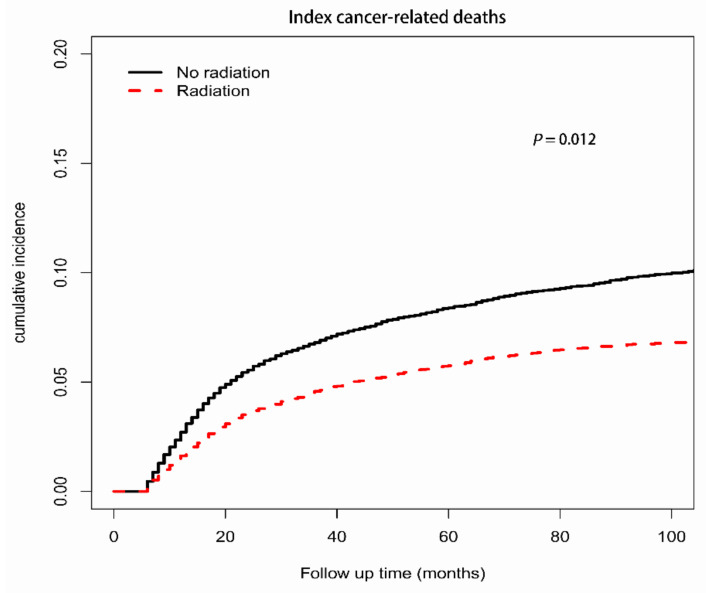
Cumulative incidence curves of death from index cancer among AYA patients, RT vs. no RT.

**Figure 2 cancers-14-05067-f002:**
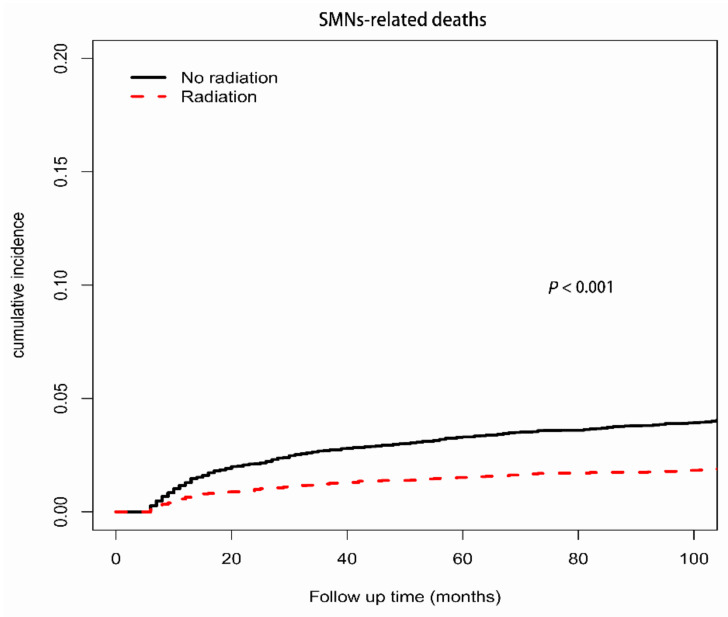
Cumulative incidence curves of death from SMNs among AYA patients, RT vs. no RT.

**Table 1 cancers-14-05067-t001:** Patient baseline demographic characteristics.

Characteristic	Overall		Radiation		No Radiation	
	n	%	n	%	n	%
Overall	29,686	100	10,708	100	18,978	100
Age, y						
15–24	9926	33.44	3944	36.83	5982	31.52
25–39	19,760	66.56	6764	63.17	12,996	68.48
Sex						
Male	16,256	54.76	5583	52.14	10,673	56.24
Female	13,430	45.24	5125	47.86	8305	43.76
Race						
White	23,648	79.66	8760	81.81	14,888	78.45
Black	3689	12.43	1050	9.81	2639	13.91
Other	2135	7.19	843	7.87	1292	6.81
Unknown	214	0.72	55	0.51	159	0.84
Era of diagnosis, year						
1992–2001	8024	27.03	3303	30.85	4721	24.88
2002–2016	21,662	72.97	7405	69.15	14,257	75.12
Marital status						
Married	10,413	35.08	3955	36.94	6458	34.03
Unmarried	18,210	61.34	6436	60.1	11,774	62.04
Unknown	1063	3.58	317	2.96	746	3.93
Ann Arbor stage						
I/II	16,463	55.46	7888	73.66	8575	45.18
III/IV	11,442	38.54	2409	22.5	9033	47.6
Unknown	1781	6	411	3.84	1370	7.22
Status						
Live	25,391	85.53	9505	88.77	15,886	83.71
Death	4295	14.47	1203	11.23	3092	16.29
Cause of death						
Index cancer	2439	56.79	712	59.19	1727	55.85
SMNs	961	22.37	222	18.45	739	23.9
Noncancer causes	895	20.84	269	22.36	626	20.25
Cardiovascular disease	208	4.84	70	5.82	138	4.46
Lymphoma subtype						
HL	17,712	59.66	7447	69.55	10,265	54.09
DLBCL	7627	25.69	2578	24.08	5049	26.6
BL	1137	3.83	124	1.16	1013	5.34
FL	1636	5.51	281	2.62	1355	7.14
MZL	297	1	57	0.53	240	1.26
MCL	85	0.29	11	0.1	74	0.39
CLL/SLL	345	1.16	27	0.25	318	1.68
PTCL	847	2.85	183	1.71	664	3.5

Abbreviations: HL, Hodgkin’s lymphoma; DLBCL, diffuse large B-cell lymphoma; MCL, mantle cell lymphoma; BL, Burkitt’s lymphoma; MZL, marginal zone lymphoma; CLL/SLL, chronic lymphocytic leukemia/small lymphocytic lymphoma; FL, follicular lymphoma; SMNs, second malignant neoplasms.

**Table 2 cancers-14-05067-t002:** Multivariable Fine–Gray regression modeling of predictors for index-cancer-related mortality, SMN-related mortality, and noncancer-disease-related mortality.

	Index-Cancer-Related Mortality		SMN-Related Mortality		Noncancer-Disease-Related Mortality	
	AHR (95% CI)	P	AHR (95% CI)	P	AHR (95% CI)	P
Treatment						
No radiation	Reference	-	Reference	-	Reference	-
Radiation	0.89 (0.81–0.97)	0.012	0.74 (0.63–0.88)	<0.001	0.77 (0.66–0.89)	<0.001
Age, y						
15–24	Reference	-	Reference	-	Reference	-
25–39	1.12 (1.01–1.24)	0.026	3.87 (3.17–4.72)	<0.001	1.47 (1.24–1.76)	<0.001
Sex						
Male	Reference	-	Reference	-	Reference	-
Female	0.87 (0.80–0.94)	<0.001	0.49 (0.42–0.57)	<0.001	0.75 (0.65–0.86)	<0.001
Year of diagnosis						
1992–2001	Reference	-	Reference	-	Reference	-
2002–2016	0.57 (0.52–0.61)	<0.001	0.45 (0.39–0.51)	<0.001	0.78 (0.68–0.9)	<0.001
Race						
White	Reference	-	Reference	-	Reference	-
Black	1.32 (1.18–1.48)	<0.001	1.86 (1.6–2.18)	<0.001	1.21 (1–1.46)	0.049
Other	1.34 (1.16–1.54)	<0.001	0.62 (0.45–0.87)	0.005	0.98 (0.75–1.3)	0.91
Unknown	0.24 (0.08–0.75)	0.014	0.22 (0.03–1.59)	0.13	0.23 (0.03–1.67)	0.15
Stage						
I/II	Reference	-	Reference	-	Reference	-
III/IV	2.01 (1.84–2.19)	<0.001	1.75 (1.52–2.03)	<0.001	1.42 (1.23–1.64)	<0.001
Unknown	1.47 (1.17–1.85)	0.001	1.74 (1.33–2.29)	<0.001	1.18 (0.82–1.68)	0.37
Marital status						
Married	Reference	-	Reference	-	Reference	-
Unmarried	1.05 (0.96–1.15)	0.31	2.61 (2.23–3.05)	<0.001	1.05 (0.91–1.23)	0.50
Unknown	0.66 (0.50–0.87)	0.0036	2.15 (1.52–3.04)	<0.001	0.91 (0.61–1.36)	0.66
Subtype						
HL	Reference	-	Reference	-	Reference	-
DLBCL	2.35 (2.14–2.57)	<0.001	2.14 (1.84–2.47)	<0.001	1.10 (0.94–1.29)	0.25
BL	2.09 (1.72–2.53)	<0.001	3.91 (3.08–4.95)	<0.001	1.12 (0.79–1.58)	0.53
FL	1.61 (1.37–1.88)	<0.001	0.54 (0.36–0.81)	0.003	1.25 (0.97–1.61)	0.078
MZL	1.27 (0.85–1.91)	0.24	0.66 (0.27–1.59)	0.35	1.25 (0.69–2.27)	0.47
MCL	3.19 (2.08–4.88)	<0.001	0.95 (0.31–2.91)	0.93	1.28 (0.48–3.39)	0.62
CLL/SLL	1.01 (0.67–1.51)	0.98	6.06 (4.5–8.16)	<0.001	2.07 (1.34–3.19)	<0.001
PTCL	3.21 (2.68–3.84)	<0.001	1.36 (0.9–2.05)	0.14	0.73 (0.45–1.19)	0.20

Abbreviations: AHR, adjusted hazard ratio; CI, confidence interval; HL, Hodgkin’s lymphoma; DLBCL, diffuse large B-cell lymphoma; MCL, mantle cell lymphoma; BL, Burkitt’s lymphoma; MZL, marginal zone lymphoma; CLL/SLL, chronic lymphocytic leukemia/small lymphocytic lymphoma; FL, follicular lymphoma; SMNs, second malignant neoplasms.

**Table 3 cancers-14-05067-t003:** Standardized mortality ratios of index-cancer-related mortality among AYA patients according to baseline characteristics.

Characteristic	Radiation	No Radiation
	SMR (95% CI)	SMR (95% CI)
Overall	534.66 * (489.9–582.41)	799.25 * (754.33–846.15)
Age, y		
15–24	806.26 * (679.11–950.31)	1258.12 * (1113.29–1416.56)
25–39	475.3 * (428.88–525.37)	720.34 * (674.27–768.74)
Sex		
Male	473.24 * (421.36–529.74)	738.42 * (687.05–792.62)
Female	648.68 * (566.33–739.63)	942.15 * (853.78–1037.18)
Race		
White	483.67 * (437.48–533.41)	724.96 * (677.42–774.95)
Black	633.48 * (493.83–800.37)	829.34 * (717.84–953.25)
Other	1274.98 * (957.81–1663.58)	2482.04 * (2057.85–2967.91)
Latency periods, m		
0–11	3221.36 * (2639.98–3892.68)	4981.89 * (4425.88–5588.43)
12–59	1256.56 * (1119.7–1405.53)	1684.83 * (1558.91–1818.21)
60–119	282.24 * (222.38–353.27)	438.62 * (375.52–509.3)
120+	82.58 * (57.84–114.33)	89.7 * (67.94–116.22)
Ann Arbor stage		
I/II	404.77 * (360.11–453.43)	577.72 * (521.73–638.09)
III/IV	937.95 * (814.81–1074.45)	1018.5 * (946.9–1094.07)
Lymphoma subtype		
HL	385.96 * (338.58–438.12)	583.55 * (529.99–641.05)
DLBCL	782.51 * (682.08–893.57)	1105.11 * (1006.2–1211.1)
BL	1704.02 * (1025.93–2661.03)	1327.53 * (1030.88–1682.96)
FL	458.3 * (283.69–700.56)	709.75 * (584.41–854.01)
MZL	379.08 * (45.91–1369.35)	755.29 * (431.72–1226.55)
MCL	967.16 * (24.49–5388.68)	2248.37 * (1161.76–3927.45)
CLL/SLL	695.97 * (143.53–2033.93)	438.53 * (264.03–684.82)
PTCL	1331.76 * (834.61–2016.3)	1544.74 * (1184.34–1980.28)

* *p* < 0.05. Abbreviations: AYA, adolescent and young adult; CI, confidence interval; HL, Hodgkin’s lymphoma; DLBCL, diffuse large B-cell lymphoma; MCL, mantle cell lymphoma; BL, Burkitt’s lymphoma; MZL, marginal zone lymphoma; CLL/SLL, chronic lymphocytic leukemia/small lymphocytic lymphoma; FL, follicular lymphoma, SMR, standardized mortality ratio.

**Table 4 cancers-14-05067-t004:** Standardized mortality ratios for nonrecurrence death after a lymphoma diagnosis among AYA patients according to cause of death.

	Radiation	No Radiation
	SMR (95% CI)	SMR (95% CI)
Second cancer death		
Oral cavity and pharynx	4.28 (0.52–15.46)	4.14 (0.85–12.09)
Digestive system	2.28 * (1.21–3.9)	2.54 * (1.59–3.85)
Respiratory system	2.67 * (1.33–4.78)	1.58 (0.76–2.91)
Bones and joints	5.46 (0.14–30.4)	11.87 * (2.45–34.68)
Soft tissue including heart	11.11 * (3.61–25.93)	1.56 (0.04–8.67)
Melanoma	1.38 (0.03–7.7)	0.97 (0.02–5.4)
Breast	1.75 (0.57–4.09)	1.85 (0.74–3.81)
Gynecologic and female genital	1.75 (0.36–5.13)	3.63 * (1.57–7.16)
Testicular and male genital	3.35 (0.08–18.67)	4.12 (0.5–14.87)
Urinary system	2.72 (0.33–9.81)	0.87 (0.02–4.87)
Brain and other nervous system	0.71 (0.02–3.97)	1.99 (0.54–5.1)
Hematopoietic #	16.52 * (10.35–25.01)	45.84 * (36.81–56.41)
Myeloma	4.42 (0.11–24.62)	5.49 (0.66–19.83)
Leukemia	19 * (11.76–29.04)	55.16 * (44.18–68.04)
ALL	12.92 * (2.66–37.75)	34.59 * (17.27–61.89)
AML	19.51 * (9.36–35.89)	22.2 * (12.69–36.06)
CLL	97 * (26.43–248.37)	632.21 * (455.64–854.57)
CML	0 (0–43.79)	23.06 * (4.76–67.4)
Noncancer deaths		
Infection	14.67 * (12.01–17.74)	33.04 * (29.85–36.48)
CVD	2.42 * (1.82–3.16)	2.8 * (2.28–3.42)
Suicide and self-inflicted injury	1.08 (0.56–1.89)	1.2 (0.72–1.88)
Diabetes mellitus	1.31 (0.36–3.35)	1.49 (0.6–3.06)
Chronic liver disease and cirrhosis	0.46 (0.06–1.65)	0.62 (0.17–1.58)
Chronic obstructive pulmonary disease and allied conditions	1.61 (0.33–4.71)	2.51 * (1.01–5.16)
Nephritis, nephrotic syndrome, and nephrosis	0.95 (0.02–5.3)	6.6 * (3.29–11.81)
Accidents and adverse effects	0.62 * (0.36–0.99)	1.02 (0.73–1.39)

* *p* < 0.05. # Hematopoietic include myeloma and leukemia. Abbreviations: AYA, adolescent and young adult; CI, confidence interval; ALL, acute lymphocytic leukemia; AML, acute myeloid leukemia; CLL, chronic lymphocytic leukemia; CML, chronic myeloid leukemia; SMR, standardized mortality ratio.

## Data Availability

The data produced and analyzed in the current study are all available in the SEER database (https://seer.cancer.gov/seerstat/ (accessed on 5 July 2020)), which is freely available to the public.

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
