# Peer review of "Role of Radiation Therapy in Mortality among Adolescents and Young Adults with Lymphoma: Differences According to Cause of Death"

_cancers, 2022, doi:10.3390/cancers14205067_

Round 1
Reviewer 1 Report
Dear authors,
a well written manuscript, which focuses on SEER database from 1992-2016. The authors reflect that RT techniques and target volume definitions changed during past five decades.
The following corrections are suggested:
-All abbreviations (e.g. SMNs in summary) primarily should be explained
-Line 36: infections are no SMNs, please correct the sentence.
-Under discussion the additional application of rituximab (since 2002) besides radiotherapy should be discussed.
Author Response
Date: Oct. 6, 2022
Dear Dr. Liu,
On behalf of all the contributing authors, I would like to express our sincere appreciation for your letter and reviewers’ constructive comments concerning our manuscript entitled "Role of radiation therapy in mortality among adolescents and young adults with lymphoma: differences according to cause of death" (Manuscript ID: cancers-1927848). These comments are all valuable and helpful for improving our manuscript. We have made modifications according to the comments. In this revised version, changes to our manuscript are highlighted using red text.
We hope that our revised manuscript will be considered for publication in Cancers. Thank you very much for your help!
Sincerely yours,
Liangshun You.
Professor
Department of Hematology, The First Affiliated Hospital, College of Medicine, Zhejiang University, #79 Qingchun Road, Hangzhou 310003, China.
Tel: 0086-571-87236717
Fax: 0086-571-87236717
E-mail: youliangshun@zju.edu.cn
Point-by-point responses are listed below.
Reviewer: 1
Comments to the Author
- All abbreviations (e.g. SMNs in summary) primarily should be explained.
Answer: Thank you for the suggestion. We have modified the corresponding text in the revised manuscript according to your comment.
- Line 36: infections are no SMNs, please correct the sentence.
Answer: Thank you for the suggestion. We apologize for the confusing grammatical and typographical errors. We have corrected the errors in the revised manuscript.
- Under discussion the additional application of rituximab (since 2002) besides radiotherapy should be discussed.
Answer: Thank you for the suggestion. We have added this argument to the discussion section of the revised manuscript to provide a detailed explanation. In the pre-rituximab era, 3 cycles of CHOP plus RT became the standard of care after the phase III SWOG S8736 study showed improved PFS and OS compared with 8 cycles of CHOP alone in localized intermediate- and high-grade non-Hodgkin's lymphoma. (Chemotherapy alone compared with chemotherapy plus radiotherapy for localized intermediate- and high-grade non-Hodgkin’s lymphoma) In modern clinical practice, chemotherapy along with rituximab have now emerged as the backbone of strategy for optimal systemic disease treatment, and RT is usually evolved to consolidate these systemic therapies to improve local control in large B-cell lymphoma. In patients who cannot tolerate chemotherapy, routine RT may spare chemotherapy; therefore, combined modality treatment (CMT), including abbreviated chemotherapy plus RT, remains an important treatment for some patients. In our analysis, SMN-related mortality was not obviously influenced by the additional application of rituximab since 2002.
We hope that the revised manuscript will meet your approval. Thank you!

Reviewer 2 Report
Thank you for the opportunity to review this well written and topical paper. My comments:
(1) Introduction: please state the research question / hypothesis more clearly
(2) Methods:
- Given the large cohort size, is a matched-pair analysis possible?
- What was the length of follow up?
- Was the length of follow-up sufficient for measurement of death by RT-induced SMN and cardiovascular events? For RT-induced mortality from SMN and cardiovascular events, it would be useful to also consider longer-term survival data (eg 20+ years).
(3) Discussion
- Eligible patients were diagnosed as recently as 2016, which, arguably, does not leave sufficient time for the true measurement of mortality from RT-induced SMN or cardiovascular events. This should be acknowledged and discussed as a limitation of this study.
- "Another cause may be that RT techniques have improved considerably in the past 50 years." Improvements over the past "50 years" are somewhat beyond the scope of this study. The authors should focus on technology changes that relate to the study period.
- "Additionally, if RT is not performed, chemotherapy must be strengthened by adding new cytotoxic drugs or increasing the current chemotherapy cycle to achieve the same effect." This statement may not always be true. Many recent protocols have attempted to de-escalate therapy with the aim to reduce risk of toxicity, whilst maintaining high cure rates.
- The paragraph on RT causing lymphopenia and infection is highly speculative, and involves extrapolating from studies of solid malignancies, higher RT doses and/or historical large RT volumes. The authors have not provided acknowledgement that the patient groups and RT techniques are vastly different, and cannot necessarily be extrapolated to patients with hematological malignancies in the modern treatment era.
- "In addition, 95% of cancer patients receiving RT will develop some form of radiodermatitis, including erythema, dry desquamation, and moist desquamation.[34] Intact skin is the first line of defense against infection." Whilst this is true overall, I find these sentences misleading and of limited relevance to the studied population: RT doses commonly used for hematological malignancies (20-36Gy to the target volume) typically result in much lower doses to the skin, and are very unlikely to break the integrity of the skin. Furthermore, the small field sizes in clinical practice are unlikely to have the same impact on cytopenias as historical extended field techniques (eg total lymphoid irradiation). I am curious to know if the authors have experience in radiotherapy for hematological malignancies?
- The SMN with the highest frequency may not equate to the SMN with the greatest risks of mortality. This should be acknowledged in the discussion. Consideration should be given to (a) the morbidity of non-fatal SMN, and (b) the temporal pattern of incidence/risk. (re: https://ascopubs.org/doi/10.1200/JCO.20.01186) Moreover, the SMN fears in current clinical practice may be based on historical treatments. eg. secondary breast cancer risk in female patients receiving RT (https://www.redjournal.org/article/S0360-3016(16)33278-3/fulltext)
- In addition to the above, the authors may wish to consider acknowledging the work of other investigators in this space: eg. https://clinicaltrials.gov/ct2/show/NCT01120353 , https://pubmed.ncbi.nlm.nih.gov/34433456/ , https://pubmed.ncbi.nlm.nih.gov/19515509/ ,
(4) References:
- I encourage the authors to reference more recent publications (eg reference 21 is from 1978)
Author Response
Point-by-point responses are listed below.
Reviewer: 2
Comments to the Author
- Introduction: please state the research question / hypothesis more clearly.
Answer: Thank you for the suggestion. We have modified the corresponding text in the revised manuscript according to your comment.
- Methods:
- Given the large cohort size, is a matched-pair analysis possible?
Answer: Thank you for the suggestion. SMRs provide the relative risk of death for patients with cancer compared to the US general residents, which was characterized adjusted by age, race, and sex to the US general population over the same time. Cause-specific mortality could not be assessed by matched-pair analysis between patients with and without RT via SMR.
- Methods:
- What was the length of follow up?
Answer: Thank you for the suggestion. The length of follow-up was from 6 months to 299 months.
- Methods:
- Was the length of follow-up sufficient for measurement of death by RT-induced SMN and cardiovascular events? For RT-induced mortality from SMN and cardiovascular events, it would be useful to also consider longer-term survival data (eg 20+ years).
Answer: Thank you for the suggestion. It is true that follow-up within the present analysis was too short to leave enough time to measure the mortality of all RT-induced SMN or cardiovascular events. The limitations have been described in detail in the discussion section.
- Discussion
- Eligible patients were diagnosed as recently as 2016, which arguably does not leave sufficient time for the true measurement of mortality from RT-induced SMN or cardiovascular events. This should be acknowledged and discussed as a limitation of this study.
Answer: Thank you for the suggestion. The limitations have been described in detail in the discussion section.
- Discussion
- "Another cause may be that RT techniques have improved considerably in the past 50 years." Improvements over the past "50 years" are somewhat beyond the scope of this study. The authors should focus on technology changes that relate to the study period.
Answer: Thank you for the suggestion. We have modified the corresponding text in the revised manuscript according to your comment.
- Discussion
- "Additionally, if RT is not performed, chemotherapy must be strengthened by adding new cytotoxic drugs or increasing the current chemotherapy cycle to achieve the same effect." This statement may not always be true. Many recent protocols have attempted to de-escalate therapy with the aim to reduce risk of toxicity, whilst maintaining high cure rates.
Answer: Thank you for the suggestion. We apologize for the confusion and errors. We have deleted the misleading statement and rephrased it as “Standard therapy for stage I and II diffuse large B-cell lymphoma consists of combined modality therapy with anthracycline-based chemotherapy, anti-CD20 antibody, and radiation therapy (RT). Curative approaches without RT typically utilize more intensive and/or protracted chemotherapy schedules.” in the revised manuscript.
- Discussion
- The paragraph on RT causing lymphopenia and infection is highly speculative, and involves extrapolating from studies of solid malignancies, higher RT doses and/or historical large RT volumes. The authors have not provided acknowledgement that the patient groups and RT techniques are vastly different, and cannot necessarily be extrapolated to patients with hematological malignancies in the modern treatment era.
Answer: Thank you for the suggestion. We apologize for the confusion and errors. We have deleted the misleading statement in the revised manuscript. It is true that there is difference in the patient group divided. Thus, when comparing the factors of death, there may be bias. However, some limitations, based on the information available in the Surveillance, Epidemiology, and End Results (SEER) database, should be taken into account when interpreting our findings. No information on RT techniques, the size of lymphoma, baseline performance status, B symptoms, or lactate dehydrogenase levels was available in the SEER database. Therefore, we could not include all the factors in the analysis. Nevertheless, we have included all available data from the SEER database in our analysis. The limitations have been described in detail in the discussion section.
- Discussion
-- "In addition, 95% of cancer patients receiving RT will develop some form of radiodermatitis, including erythema, dry desquamation, and moist desquamation.[34] Intact skin is the first line of defense against infection." Whilst this is true overall, I find these sentences misleading and of limited relevance to the studied population: RT doses commonly used for hematological malignancies (20-36Gy to the target volume) typically result in much lower doses to the skin, and are very unlikely to break the integrity of the skin. Furthermore, the small field sizes in clinical practice are unlikely to have the same impact on cytopenias as historical extended field techniques (eg total lymphoid irradiation). I am curious to know if the authors have experience in radiotherapy for hematological malignancies?
Answer: Thank you for the suggestion. We apologize for the confusion and errors. We have deleted the misleading statement in the revised manuscript.
- Discussion
- The SMN with the highest frequency may not equate to the SMN with the greatest risks of mortality. This should be acknowledged in the discussion. Consideration should be given to (a) the morbidity of non-fatal SMN, and (b) the temporal pattern of incidence/risk. (re: https://ascopubs.org/doi/10.1200/JCO.20.01186) Moreover, the SMN fears in current clinical practice may be based on historical treatments. eg. secondary breast cancer risk in female patients receiving RT (https://www.redjournal.org/article/S0360-3016(16)33278-3/fulltext).
Answer: Thank you for the suggestion. We have added the following arguments: “The SMN with the highest frequency may not equate to the SMN with the greatest risks of mortality.” “Moreover, the SMN fears in current clinical practice may be based on historical treatments., e.g., secondary breast cancer risk in female patients receiving RT” to the discussion section of the revised manuscript. It is true that the morbidity of nonfatal SMN and the temporal pattern of incidence/risk included in the revised manuscript, we can know how the incidence of common secondary tumors is caused by radiotherapy. This is more meaningful than simply statistical SMN mortality rate. However, some limitations, based on the information available in the SEER database, should be taken into account when interpreting our findings. The limitations have been described in detail in the discussion section.
- Discussion
- In addition to the above, the authors may wish to consider acknowledging the work of other investigators in this space: eg. https://clinicaltrials.gov/ct2/show/NCT01120353, https://pubmed.ncbi.nlm.nih.gov/34433456/, https://pubmed.ncbi.nlm.nih.gov/19515509/.
Answer: Thank you for the suggestion. The work of drawing on other investigators in this field has made this article a further increase. We have modified the corresponding text in the revised manuscript according to your comment.
- References:
- I encourage the authors to reference more recent publications (eg reference 21 is from 1978).
Answer: Thank you for the suggestion. We have modified the corresponding text in the revised manuscript according to your comments.
We hope that the revised manuscript will meet your approval. Thank you!